# Metabolomic response of *Euglena gracilis* and its bleached mutant strain to light

Qing Shao[1], Lang Hu[2], Huan Qin[1], Yerong Liu[1], Xing Tang[1], Anping Lei[1]*, Jiangxin Wang[1]*

1 Shenzhen Key Laboratory of Marine Bioresource and Eco-environmental Science, Shenzhen Engineering Laboratory for Marine Algal Biotechnology, Guangdong Provincial Key Laboratory for Plant Epigenetics, College of Life Sciences and Oceanography, Shenzhen University, Shenzhen, China, 2 College of Life Sciences and Technology, Hubei Engineering University, Xiaogan, China

☯ These authors contributed equally to this work.
* bioaplei@szu.edu.cn (AL); jxwang@szu.edu.cn (JW)

**Data Availability Statement:** All relevant data are within the manuscript and its Supporting Information files.

**Funding:** This work was supported by National Natural Science Foundation of China (31670116 to AL), the Guangdong Innovation Research Team

## Abstract

*Euglena*, a new superfood on the market, is a nutrient-rich, green single-celled microorganism that features the characteristics of both plant and animal. When cultivated under different conditions, *Euglena* produces different bioactive nutrients. Interestingly, *Euglena* is the only known microorganism whose chloroplasts are easy to lose under stress and become permanently bleached. We applied gas chromatography-mass spectrometry (GC-MS) to determine the metabolomes of wild-type (WT) *Euglena gracilis* and its bleached mutant OflB2 under light stimulation. We found a significant metabolic difference between WT and OflB2 cells in response to light. An increase of membrane components (phospholipids and acylamides) was observed in WT, while a decrease of some stimulant metabolites was detected in OflB2. These metabolomic changes after light stimulation are of great significance to the development of *Euglena* chloroplasts and their communications with the nucleus.

## Introduction

*Euglena* is a single-celled eukaryotic microalga and shares characteristics of both animals and plants. It has no cell wall but contains a photosensitive eyespot structure and a flagellum. Among its bounty of nutrients are 14 types of Vitamins, 9 minerals, 18 amino acids, 11 unsaturated fatty acids, and 7 others like chlorophyll and paramylon (β-glucan). Different cultivations of *Euglena* result in a significant diversity of nutrients inside the cells [1]. The *Euglena* cells contain fully developed chloroplasts that they can grow both autotrophically under light and heterotrophically by absorbing organic matters from the environment [2]. The chloroplasts of *Euglena* are believed originated from the prochloron by endosymbiosis. Unlike the chloroplasts of other microalgae or higher-level plants, the chloroplasts of *Euglena* have poor stability and are prone to lose part of the chloroplast genome under stress such as antibiotics [2]. The *Euglena* cells losing part of their photosynthesis-related genes result in bleached mutants that can stably grow and multiply in heterotrophic media.

Fund (2014ZT05S078 to JW), and the Shenzhen Grant Plan for Science & Technology (JCYJ20160308095910917, JCYJ20170818100339597, both to JW). They are used for the design of the study, data collection, data analysis and manuscript writing, respectively. The funders had no role in study design, data collection and analysis, decision to publish, or preparation of the manuscript.

**Competing interests:** The authors have declared that no competing interests exist.

Chloroplasts are differentiated from proplastids without internal structure. In different environments or tissues, proplastids can develop into plastid structures with different functions, such as leukoplastid, amyloplast, chromoplast and chloroplast. Light is a prerequisite for *Euglena* chlorophyll biosynthesis, and normally chloroplasts cannot develop in the dark. In the process of chloroplast assembly, photopigments regulate the synthesis of chloroplast-related proteins under light stimulation [3]. Some of chloroplast-related proteins are encoded by nuclear genes and are transported from the cytoplasm to the protoplasts by transport peptides. These proteins help improve the structure and functional development of the chloroplasts. For example, the increases of chlorophyll, the formation of thylakoids and the development of some electron transport chain complexes all depend on these proteins [3]. In addition, cell division and chloroplast division are kept in synchronization, resulting in the metabolic state of the cell changes during the chloroplast development [4].

Metabolomics refers to the study of metabolites from the various metabolic pathways of a certain organism. Qualitative and quantitative metabolomics can be used to reveal the changes of metabolic state, detect the biomarkers, and elucidate the related metabolic pathways and mechanisms of the organism responding to environmental stimuli [5–10]. The metabolomics play an important role in systems biology, and it benefits drug development [5], toxicological study [6], and disease diagnosis [7]. The most common methods used in metabolomics include nuclear magnetic resonance spectroscopy (NMR), gas chromatography-mass spectrometry (GC-MS), liquid chromatography coupled with mass spectrometry (LC-MS). The application of metabolomics in microalgal has increased year by year. For example, the metabolome change induced by stress has been investigated in *Ectocarpus siliculosu*, *Chlorella vulgaris* and *Pseudochoricystis ellipsoidea* [11–13]. In *Euglena*, although there are countless reports to investigate the chloroplast development at the morphological levels [14–16], metabolomics studies are hardly seen [17]. GC-MS can identify hundreds of compounds at the same time, and has the advantages of high sensitivity, high resolution, and database support. We have used this separation and identification technology to detect the changes of metabolites of algae under light stimulation [17, 18].

In order to uncover the related mechanisms of chloroplast development under light stimulation in *Euglena*, we applied light stimulation to two strains of *Euglena gracilis*–wild type (WT) and bleached mutant OflB2 obtained by oxflaxicin treatment and performed GC-MS to detect the metabolome. This study reveals the physiological changes related to chloroplast development at the metabolomic level and provides a theoretical basis for further studies of the biosynthesis of chloroplast and its retrograde regulation with the nucleus.

## Materials and methods

### Algae cultivation and light stimulation

Wide type *E. gracilis* CCAP 1224/5Z was purchased from Culture Collection of Algae and Protozoa (CCAP) and maintained in our laboratory. The bleached mutant OflB2 permanently lost functional chloroplasts and was obtained by treating the WT of *E. gracilis* CCAP 1224/5Z with ofloxacin [2]. WT *E. gracilis* and the bleached mutant OflB2 were all cultured in Hetero medium (pH 3) [19] at the temperature of 23 ± 1 ˚C.

### Absorption spectrum of the pigments

$10^6$ cells were mixed with 1 mL of 95% ethanol and incubated at 4 ˚C overnight. After centrifugation at 8000 g for 5 min, the absorption spectrum of supernatant was recorded by spectrometer. The detection wavelength was set to 300 nm—700 nm, and the scanning speed was 20 nm $min^{-1}$.

## Metabolome detection by GC-MS

To investigate the influence of light stimulation on the metabolite compositions of WT and OflB2, WT and OflB2 were first cultured under dark. After 6 days of the dark cultivation (samples entered into the platform stage), some cells were collected (used for 0 h sample under light) and the remaining cells were light-stimulated at the light intensity of $65 \pm 5$ μmol m$^{-2}$ s$^{-1}$. The light-treated samples were collected at 4 h, 12 h, 24 h, 48 h, and 72 h. Each sample had three parallels. The method for extraction of the metabolome was described in the previously published literatures [17, 18]. The metabolome was examined using the GC-7890A-MS-5975C gas chromatography-mass spectrometry system (Agilent Technologies, Inc.). The GC column was HP-5MS capillary column (Part No.: 19091S-433), which was 30.0 m (length) $\times$ 250 μm (inside diameter) $\times$ 0.25 μm (film thickness). The GC inlet temperature was set to 230 ˚C. Helium with purity higher than 99.999% was used as the carrier gas. 2 μL sample was injectied in splitless mode. The temperature procedure of the column oven was set as follows: 45 ˚C incubated for 2 min, raised to 280 ˚C at a rate of 5 ˚C min$^{-1}$, 280˚C incubated for 2 min. The transmission line temperature, ion source temperature and MS quadrupole were set to 290 ˚C, 230 ˚C and 150 ˚C, respectively. Metabolites retained in capillary column for at least 10 minutes. Full scan mode was used, and the charge-to-mass ratio detection range was set to 50–550.

The raw data from the GC-MS analysis were analyzed using the Agilent Automated Mass Spectrometry Deconvolution and Recognition System (AMDIS). For qualitative analysis of metabolites, the mass fragmentation spectrums were matched to the Fiehn database and the database of National Institute of Standards and Technology (NIST). The relative amount of each metabolite is the ratio of the peak area of the metabolite to that of the internal standard. Finally, the relative amount of metabolite was normalized by the number of cells.

Principal Component Analysis (PCA) was conducted to show the relationship between samples. In order to reveal the changes in expression patterns of metabolites, Cluster Analysis (CA) analysis was conducted and heatmap was drawn. In the heatmap, the raw data were $\log_{10}$-transformed to better show the differences of the expression pattern [20].

## Results

### Growth of WT *E. gracilis* and OflB2

The growths of WT *E. gracilis* and OflB2 under dark or light were shown in Fig 1. No matter under light or dark, the growth of WT *E. gracilis* was always better than that of OflB2. The cell density of WT was higher than that of OflB2 from day 3 to day 9. From day 6 to day 9, the effect of light on the growth was more obvious for WT *E. gracilis* than for OflB2. Under dark the highest cell density of WT was achieved at day 7 while under light it was achieve at day 9. For OflB2, there were little differences in growth between under dark and under light.

### Pigment compositions in *E. gracilis* and OflB2

In order to investigate the differences in pigment composition between WT *E. gracilis* and OflB2, the absorption spectra of pigments were recorded (Fig 2). There were absorption peaks at 433 nm and 665 nm in WT *E. gracilis*, but no such absorption peaks were found in OflB2, indicating that WT has pigments while OflB2 contains no pigments.

### GC-MS analysis

A total of 36 samples (2 strains at 6 time points with 3 biological replicates) were collected for metabolome detection. After filtering the metabolites with low quality scores (below 70), 78 and 61 metabolites were detected in WT *E. gracilis* and OflB2, respectively (see the S1 Table).

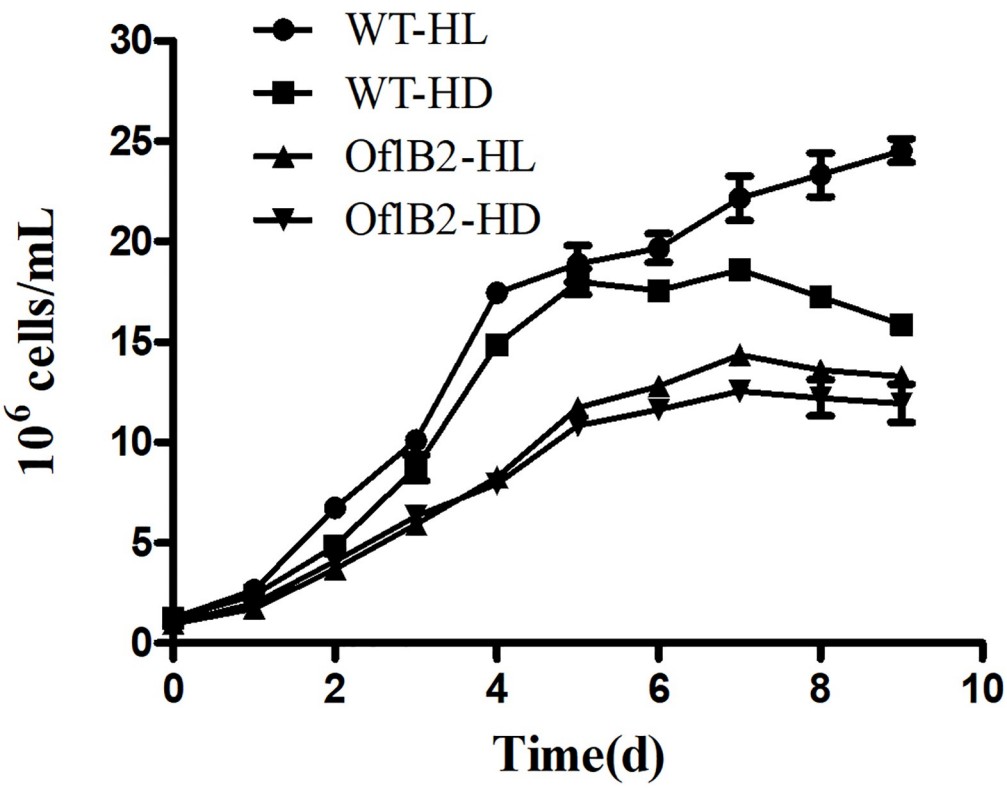

**Fig 1. The growth of WT *E. gracilis* and OflB2 in Hetero medium under light (HL) and dark (HD) conditions.**

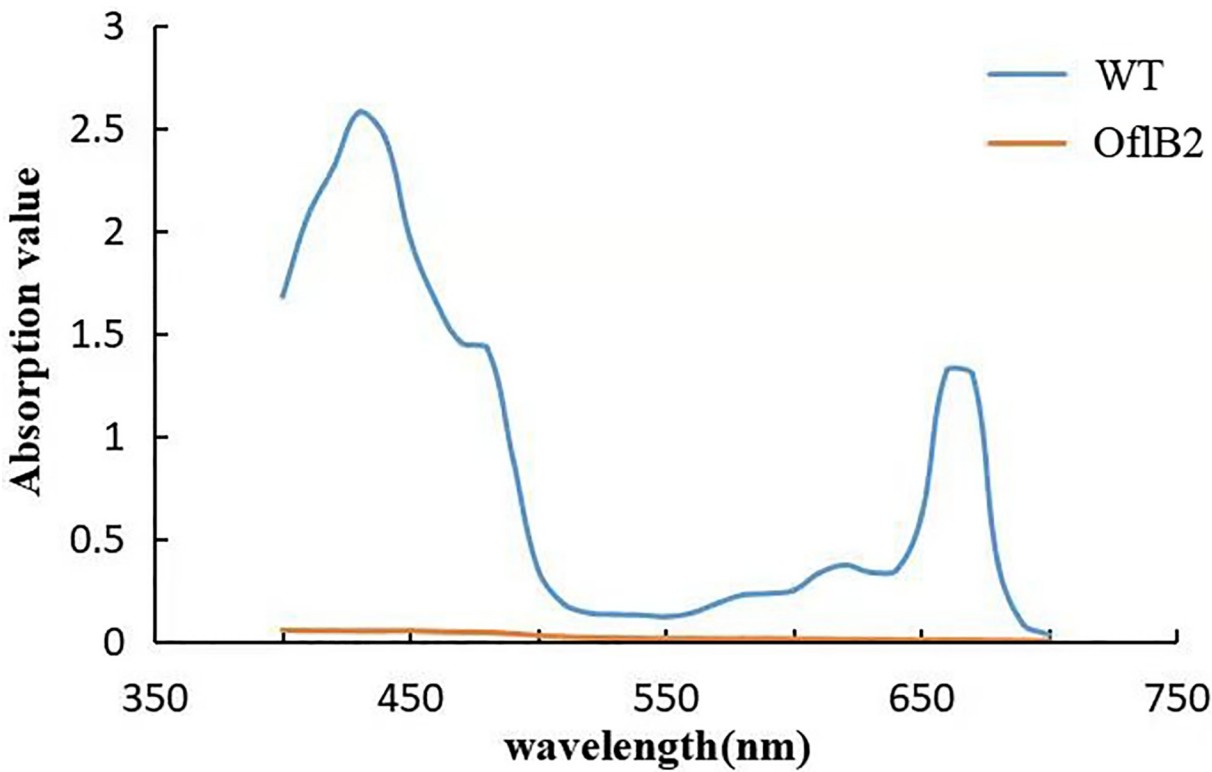

**Fig 2. The absorption spectrum of the pigments in WT *E. gracilis* and OflB2.**

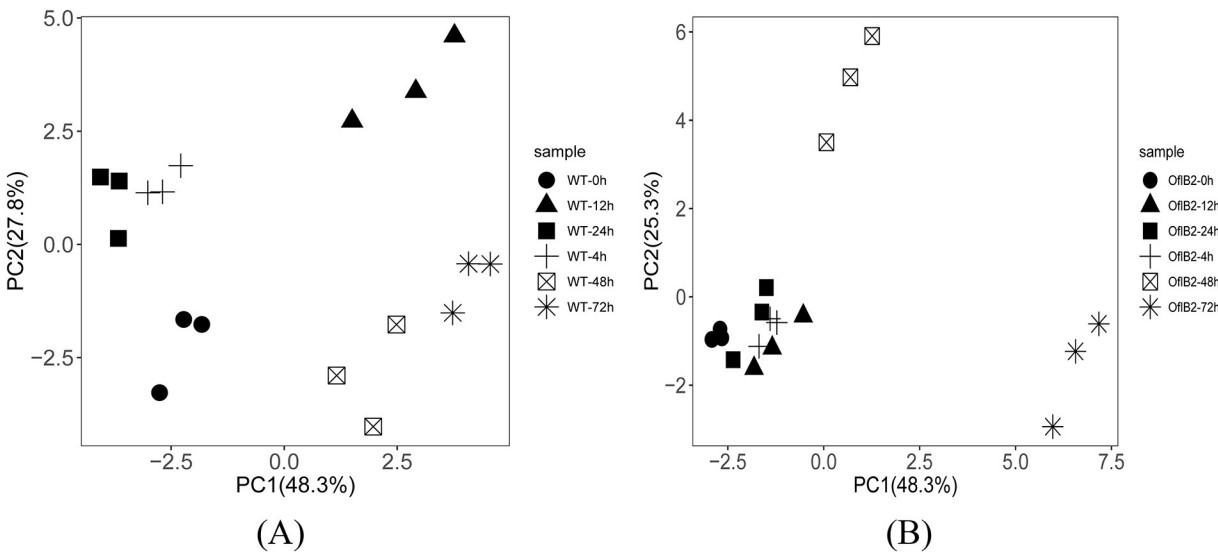

**Fig 3. PCA analysis of WT *E. gracilis* (a) and OflB2 (b) against light treatment.**

## Principal Component Analysis (PCA)

PCA was conducted to see whether there were significant metabolomic differences between WT *E. gracilis* and OflB2 after light stimulation. The results were shown in Fig 3. The cumulative contribution of PC1 and PC2 was 76.1% for WT *E. gracilis*, and 73.6% for OflB2, indicating that this first two principal components account for most of the variations. For WT *E. gracilis*, samples with different light durations were separated from each other (Fig 3), suggesting that the metabolome of WT *E. gracilis* changes with light duration and WT *E. gracilis* cells adapt to changing light stimulation through necessary physiological and biochemical changes. In OflB2 (Fig 3), three clusters (samples with less than 24 h of light stimulation, samples with 48 h of light stimulation and samples with 72 h of light stimulation) can be found, suggesting that little changes in metabolic pathway occurred in the first 24 h of light stimulation and a certain light duration of more than 24 h can induce the metabolome change in the bleached OflB2 mutant cells. It also cannot be ruled out that the change in metabolite components after 24 h may be caused by other factors such as cell aging.

## Cluster analysis

Cluster analysis was carried out to uncover the metabolomic changes after light stimulation in WT *E. gracilis* and OflB2. In WT *E. gracilis*, light stimulation results in upregulation of some metabolites (Fig 4). These metabolites includes amino acids and lipid metabolites, such as phosphonolipid (phosphoric acid, bis(trimethylsilyl) monomethyl ester), benzoic ester (phthalic acid, 6-ethyloct-3-yl 2-ethylhexyl ester), and glycine ester (glycine, N,N-bis(trimethylsilyl)-, trimethylsilyl ester), as well as the energy substance such as saccharide (d-Mannose). Light stimulation also changed the intracellular metabolism in OflB2 (Fig 4) and some stimulant metabolites decreased after light stimulation.

## Discussion

Since WT *Euglena* could grow to high cell density in either light or dark conditions, we proposed that the fully developed chloroplasts would not be helpful, even not a burden, to cells

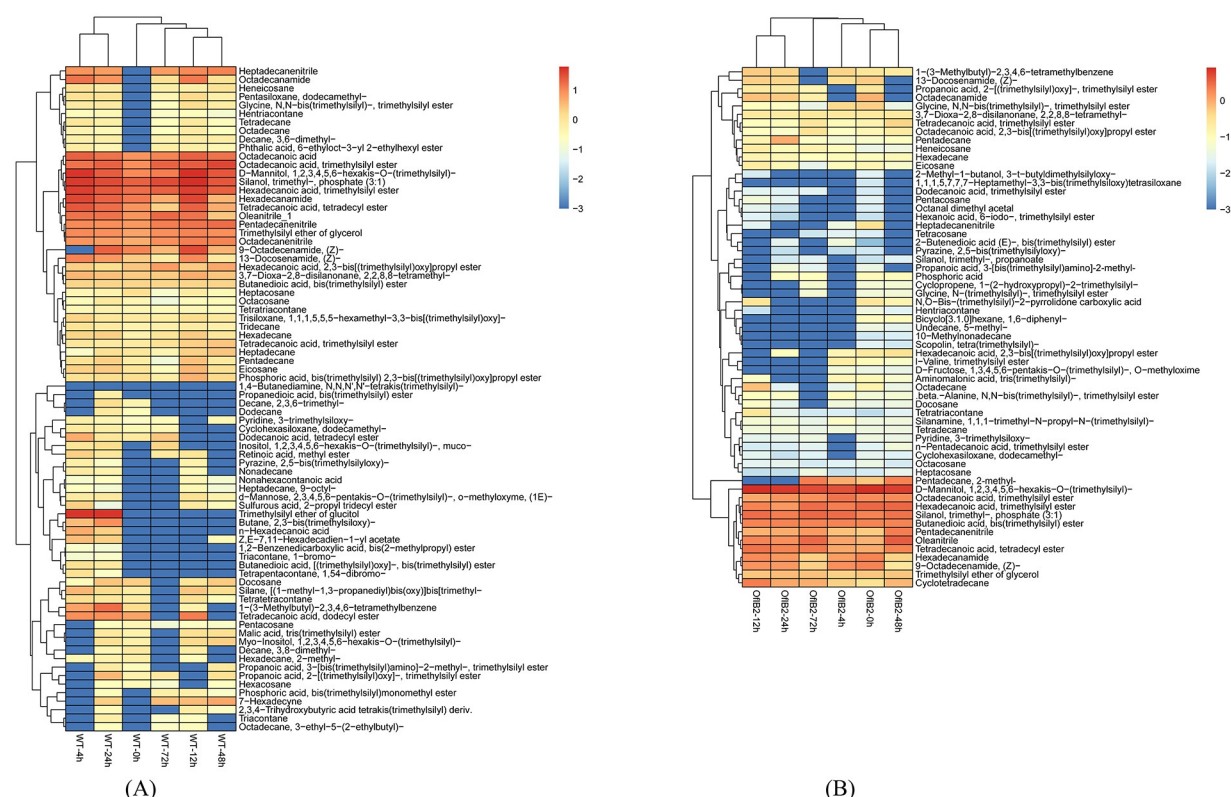

**Fig 4. Heat map of WT *E. gracilis* (a) and OflB2 (b) metabolites after light stimulation.**

grown in the dark, where photosynthetic function is not needed. In the dark WT *Euglena* cannot photosynthesize and the chloroplasts inside cells may be a burden for cells. Without chloroplasts in the dark, the mutant cells were expected to grow better than WT since they have no material and energy waste for chloroplasts. Thus, the bleached mutants only growing heterotrophically may show benefit for better growth compared to WT under the same condition. However, unexpectedly, the growth of the OflB2 mutant was much slower than that of WT in the dark, probably because the OflB2 mutant has lost chloroplast and partial plastid genome, which may affect the growth and reproduction of OflB2 mutant in the dark [2].

As early as in the 1970–80s, there have been reports about the photo-induced synthesis of enzymes for fixing carbon dioxide during early chloroplast development stage [21], demonstrating that Rubisco increased its activity during the 12 h after the pre-light stimulation of the *Euglena* cells [22, 23]. It has also been reported that under the exposure to light, mitochondrial surface area, mitochondrial lipid levels, cytochromes were found decreased [24], and cyclophorase, such as the delta-aminolevulinic acid (pronounce) synthase (ALA synthase) [25, 26] and mitochondrial elongation factors G, was also down-regulated [27], while phosphoenolpyruvate carboxylase slightly increased [28, 29]. Under the exposure to light, the 44 mitochondrial proteins on the whole-cell 2-D gel were detected to reduce by 31 in relative quantity, about 70% in WT *Euglena* [30, 31]. Visible light-induced selectivity alters the rate of specific mitochondrial protein synthesis. In addition, after exposure to light, a slightly lower cytosolic aldolase activity was found [32], similar down-regulated activity was also noticed in phosphofructokinase [32], phosphoenolpyruvate carboxykinase [33], NAD-glyceraldehyde-3-phosphate dehydrogenase [21], and malic enzyme [34]. It was concluded that, compared with

OflB2, WT *Euglena* will experience a change in nutrition mode after light stimulation, gradually changing from heterotrophic to photoautotrophic state. This process mainly includes the synthesis of chlorophyll, chloroplast assembly, and changes in energy supplies of mitochondrial. The changes of enzyme levels caused by various physiological and biochemical reactions in the cytoplasm will eventually lead to changes in some major metabolite components.

We reported the relatively significant differences in either response patterns or metabolite components between WT and OflB2 under light stimulation in this study. The individual significantly changed components are somehow involved in the accumulation of chlorophyll, formation of thylakoid structures [35], and synthesis of photosynthesis-related proteins and others in the process of chloroplast development. It was once reported that the biosynthesis of phosphatidylglycerol (PG) is the basis of embryonic development as well as normal membrane structure of chloroplasts and mitochondria [36]. It was stated that galactolipids, monogalacto-syl-diacylglycerol (MGDG), and digalactosyl-diacylglycerol (DGDG), major lipids in the thylakoid membrane required for its development, could be essential for maintaining optimal efficiency in photosynthesis [37, 38]. X-ray crystallographic studies of photosystem I (PS I) and photosystem II (PS II) also indicated that thylakoid lipids were contained in the crystal structures of both compounds, and that PG and MGDG were both found near the reaction center of PS I and PS II [39, 40]. In addition, we found that compared with the control group WT-0h, WT-4h had a different metabolite composition, further suggesting that cell metabolism of WT *E. gracilis* changes very quickly, 4 h as a short period of time after light stimulation. This change is significantly represented by the increase of some important metabolites. For instance, the increase of phosphonolipid may be related to the increase of membrane components; the formation of thylakoids in chloroplasts will be accelerated after light stimulation; and the increase of other amino acids and amides is associated with the synthesis of some photosynthetic proteins. Therefore, it is safe to conclude that the changes in the metabolite components and contents of WT *Euglena* after light treatment are mostly due to the accumulation of substances and energy for photosynthesis.

Since WT *E. gracilis* performs photosynthesis, it can be understood that the enhancement of metabolic activity is to prepare some substances for photosynthesis. However, bleached mutant OflB2 did not contain functional chloroplasts and thus performed no photosynthesis, and we infer that light stimulation may reduce the transcription of certain genes and result in the decrease of the levels of major metabolites in the cells.

Combined with other Omic data analyses, such as genomic, transcriptomic, and proteomic data (unpublished) conducted in WT *E. gracilis* and bleached mutants, we propose that the regulation of light induced photoreactivity in *Euglena* is more likely to occur at the post-transcriptional level. Metabolomic analysis showed that metabolome of WT *E. gracilis* under light stimulation is majorly represented by its relation with the increase of membrane components. For example, light promotes the formation of thylakoids in chloroplasts; the increase of other amino acids and amides is associated with the synthesis of some photosynthetic proteins. However, under light stimulation, metabolome of the bleached OflB2 is characterized by the down-regulation of some metabolite components. It is proposed that light stimulation slows down the metabolism of the bleached *Euglena* cells to a certain extent, making it resemble a semi-dormant state in a short period of time, thereby reducing the damage of light stress against cells without any functional chloroplasts. In summary, the metabolomic analyses of WT *E. gracilis* and bleached mutant OflB2 showed that metabolite component changes occur in both strains after light stimulation. In the WT *E. gracilis*, it is mainly represented by the increase of some metabolites such as phosphoric ester involved in membrane synthesis and various amino acids in protein synthesis for photosynthesis. In contrast, in the bleached OflB2, metabolomes were mainly characterized by the decrease of some secondary metabolites. In

addition, cells of the two strains showed different response patterns to light stimulation: WT *E. gracilis* changed more rapidly than the bleached cells. Therefore, there is a significant difference in the response mechanism between the two strains to light stimulation. We propose that the main cause in WT *E. gracilis* is to switch from heterotrophy to autotrophy, so as to prepare itself for chloroplast development and photosynthesis. However, the bleached mutant OflB2 cells are at the heterotrophic state, and light stimulation may work as a stress instead of a chloroplast development. As a result, its cells adapt to the new environment by changing the metabolomic status.

## Supporting information

**S1 Table. Intracellular metabolites detected with GC-MS in WT *E. gracilis* and OflB2.** (XLSX)

## Author Contributions

**Conceptualization:** Jiangxin Wang.

**Data curation:** Lang Hu.

**Formal analysis:** Xing Tang.

**Investigation:** Qing Shao, Lang Hu.

**Methodology:** Qing Shao.

**Project administration:** Jiangxin Wang.

**Validation:** Lang Hu.

**Writing – original draft:** Lang Hu, Huan Qin, Anping Lei.

**Writing – review & editing:** Lang Hu, Huan Qin, Yerong Liu, Anping Lei, Jiangxin Wang.

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
