## [Decision Letter · Decision Letter 0]

15 Jul 2019

PONE-D-19-15939

Metabolomic response of Euglena gracilis and its bleached mutant strain to light

PLOS ONE

Dear Dr. Wang,

Thank you for submitting your manuscript to PLOS ONE. After careful consideration, we feel that it has merit but does not fully meet PLOS ONE’s publication criteria as it currently stands. Therefore, we invite you to submit a revised version of the manuscript that addresses the points raised during the review process.

We would appreciate receiving your revised manuscript by Aug 29 2019 11:59PM. To enhance the reproducibility of your results, we recommend that if applicable you deposit your laboratory protocols in protocols.io, where a protocol can be assigned its own identifier (DOI) such that it can be cited independently in the future. For instructions see: http://journals.plos.org/plosone/s/submission-guidelines#loc-laboratory-protocols

We look forward to receiving your revised manuscript.

Kind regards,

Haitao Shi

Academic Editor

PLOS ONE

Journal Requirements:

1. Thank you for including the following funding information within the acknowledgements section of your manuscript; "This work was supported by National Natural Science Foundation of China (31670116), the Guangdong Innovation Research Team Fund (2014ZT05S078), and the Shenzhen Grant Plan for Science & Technology (JCYJ20160308095910917, JCYJ20170818100339597), used for the design of the study, data collection, data analysis and manuscript writing. The authors gratefully acknowledge the supports from the Instrumental Analysis Center of Shenzhen University (Xili Campus)."

"NO - Include this sentence at the end of your statement: The funders had no role in study design, data collection and analysis, decision to publish, or preparation of the manuscript."

2. Please amend your list of authors on the manuscript to ensure that each author is linked to an affiliation. Authors’ affiliations should reflect the institution where the work was done (if authors moved subsequently, you can also list the new affiliation stating “current affiliation:….” as necessary).

Reviewers' comments:

Reviewer's Responses to Questions

**Comments to the Author**

1. Is the manuscript technically sound, and do the data support the conclusions?

Reviewer #1: Yes

Reviewer #2: Partly

2. Has the statistical analysis been performed appropriately and rigorously? 

Reviewer #1: Yes

Reviewer #2: Yes

3. Have the authors made all data underlying the findings in their manuscript fully available?

Reviewer #1: Yes

Reviewer #2: No

4. Is the manuscript presented in an intelligible fashion and written in standard English?

Reviewer #1: Yes

Reviewer #2: No

5. Review Comments to the Author

Reviewer #1: This is an interesting study of a unique type of algae. The experimental design is sound and the results are useful. There are a few points needs to be amended or addressed.

The last paragraph of the introduction, no result should be presented or discussed in the introduction.

Line 103, what does it mean "The solvent was delayed by 10 min"?

Line 191, why the low growth of growth of the OflB2 mutant is unexpected? this needs better explanation.

It is hard for readers to understand when conclusion is made from unpublished data. Suggest to delete or modify the 2nd paragraph of the summary.

Reviewer #2: The manuscript applied gas chromatography-mass spectrometry (GC-MS) to determine the metabolomes of wild-type (WT) Euglena gracilis and its bleached mutant OflB2 under light stimulation. These metabolomic changes after light stimulation are of great significance to the development of Euglena chloroplasts and their communications with the nucleus. However, The manuscript is relatively simple, the authors should consider the following points:

1. Figure 1 cannot be found.

2. Morphological changes of chlorophyll and other physiological indicators should be observed

3. Add GC-MS Analysis of Raw Data Graph.

3. It is suggested that genomics, transcriptome and proteomics data be published together to better reveal the mechanism of light effects on them.

4.There are some grammatical errors in the manuscript,please modify the language further.

6. PLOS authors have the option to publish the peer review history of their article (what does this mean?). If published, this will include your full peer review and any attached files.

Reviewer #1: No

Reviewer #2: No

---

## [Author Response · Author response to Decision Letter 0]

22 Oct 2019

Reviewer #1

1.Questions: The last paragraph of the introduction, no result should be presented or discussed in the introduction.

Response: Thanks for your constructive advices. We have moved the sentence “the relatively significant differences in either response time or metabolite components between WT and OflB2 under light stimulation” from introduction into discussion (lines 189-190).

2.Questions: Line 103, what does it mean "The solvent was delayed by 10 min"?

Response: Sorry for the confusion caused by the unclear description. We changed the "The solvent was delayed by 10 min" to "Metabolites retained in capillary column for at least 10 minutes"(lines 95-96).

3.Questions: Line 191, why the low growth of growth of the OflB2 mutant is unexpected? this needs better explanation.

Response: Since WT Euglena could grow heterotrophically and autotrophically under different culture conditions, and in the dark WT Euglena cannot photosynthesize and the chloroplasts inside cells may be a burden for cells. Without chloroplasts in the dark, the mutant cells were expected to grow better than WT since they have no material and energy waste for chloroplasts. We have added this paragraph into the revised manuscript (Line 159-162). 

4.Questions: It is hard for readers to understand when conclusion is made from unpublished data. Suggest to delete or modify the 2nd paragraph of the summary.

Response: Thanks for useful suggestions. We have deleted the conclusion based on unpublished data.

Reviewer #2 

1.Questions: Figure 1 cannot be found.

Response: Sorry for low mistake by the first time submissioner. We have added figures in the revised version and this time we promise every figure are included. 

2.Questions: Morphological changes of chlorophyll and other physiological indicators should be observed

Response: in the new version we added and analyzed the differences of pigments between wild type and the bleached mutant in lines 121-125

3.Questions: Add GC-MS Analysis of Raw Data Graph.

Response: we have added a supplementary material containing the raw analysis data of GC-MS in the newly revised version. 

4.Questions: It is suggested that genomics, transcriptome and proteomics data be published together to better reveal the mechanism of light effects on them.

Response: thanks for constructive advice. We agree completely with you. We have the RNA-seq and proteomics data but still wait for the ongoing Euglena genome assemble, to make a better annotation and analysis. 

5.Questions: There are some grammatical errors in the manuscript, please modify the language further.

Response: we have corrected the grammatical errors as possible as we can. Please check for the details in the revised manuscript.

Academic Editor：

We will only publish funding information present in the Funding Statement section of the online submission form.

Please amend your list of authors on the manuscript to ensure that each author is linked to an affiliation.

Response: we have deleted the funding information from the manuscript text, as request. We amended the list of authors and ensure that each author has only one major affiliation.

---

## [Editor Report · Decision Letter 1]

25 Oct 2019

Metabolomic response of Euglena gracilis and its bleached mutant strain to light

PONE-D-19-15939R1

Dear Dr. Wang,

We are pleased to inform you that your manuscript has been judged scientifically suitable for publication and will be formally accepted for publication once it complies with all outstanding technical requirements.

With kind regards,

Haitao Shi

Academic Editor

PLOS ONE
---

## [Editor Report · Acceptance letter]

31 Oct 2019

PONE-D-19-15939R1 

Metabolomic response of Euglena gracilis and its bleached mutant strain to light 

Dear Dr. Wang:

I am pleased to inform you that your manuscript has been deemed suitable for publication in PLOS ONE. Congratulations! Your manuscript is now with our production department. 

With kind regards,

on behalf of

Dr. Haitao Shi 

Academic Editor

PLOS ONE